# Rapid Detection of Cleanliness on Direct Bonded Copper Substrate by Using UV Hyperspectral Imaging

**DOI:** 10.3390/s24144680

**Published:** 2024-07-19

**Authors:** Mona Knoblich, Mohammad Al Ktash, Frank Wackenhut, Tim Englert, Jan Stiedl, Hilmar Wittel, Simon Green, Timo Jacob, Barbara Boldrini, Edwin Ostertag, Karsten Rebner, Marc Brecht

**Affiliations:** 1Center of Process Analysis and Technology (PA&T), School of Life Sciences, Reutlingen University, Alteburgstraße 150, 72762 Reutlingen, Germany; mona.knoblich@reutlingen-university.de (M.K.); mohammad.alktash@reutlingen-university.de (M.A.K.); frank.wackenhut@reutlingen-university.de (F.W.); barbara.boldrini@reutlingen-university.de (B.B.); edwin.ostertag@rpt.bwl.de (E.O.); karsten.rebner@reutlingen-university.de (K.R.); 2Institute of Physical and Theoretical Chemistry, Eberhard Karls University Tübingen, Auf der Morgenstelle 18, 72076 Tübingen, Germany; 3Robert Bosch GmbH, Automotive Electronics, Tübingerstraße 123, 72762 Reutlingen, Germany; tim.englert2@de.bosch.com (T.E.); jan.stiedl@de.bosch.com (J.S.); simon.green@de.bosch.com (S.G.); 4Center of Physics, Reutlingen University, Alteburgstraße 150, 72762 Reutlingen, Germany; hilmar.wittel@reutlingen-university.de; 5Institute of Electrochemistry, Ulm University, Albert-Einstein-Allee 47, 89081 Ulm, Germany; timo.jacob@uni-ulm.de

**Keywords:** hyperspectral imaging, pushbroom, UV spectroscopy, principal component analysis, partial least squares regression, electrical copper, cleanliness, soldering

## Abstract

In the manufacturing process of electrical devices, ensuring the cleanliness of technical surfaces, such as direct bonded copper substrates, is crucial. An in-line monitoring system for quality checking must provide sufficiently resolved lateral data in a short time. UV hyperspectral imaging is a promising in-line method for rapid, contactless, and large-scale detection of contamination; thus, UV hyperspectral imaging (225–400 nm) was utilized to characterize the cleanliness of direct bonded copper in a non-destructive way. In total, 11 levels of cleanliness were prepared, and a total of 44 samples were measured to develop multivariate models for characterizing and predicting the cleanliness levels. The setup included a pushbroom imager, a deuterium lamp, and a conveyor belt for laterally resolved measurements of copper surfaces. A principal component analysis (PCA) model effectively differentiated among the sample types based on the first two principal components with approximately 100.0% explained variance. A partial least squares regression (PLS-R) model to determine the optimal sonication time showed reliable performance, with *R*^2^_cv_ = 0.928 and RMSECV = 0.849. This model was able to predict the cleanliness of each pixel in a testing sample set, exemplifying a step in the manufacturing process of direct bonded copper substrates. Combined with multivariate data modeling, the in-line UV prototype system demonstrates a significant potential for further advancement towards its application in real-world, large-scale processes.

## 1. Introduction

Copper is essential in the industrial environment due to its exceptional properties, including high electrical and thermal conductivity, antimicrobial activity, and corrosion resistance [1,2,3]. These properties make it widely used in the semiconductor industry for integrated circuit packaging and wire bonding [1,4]. However, copper surfaces can become contaminated and react with organic materials and oxygen during soldering, glueing, or wire bonding processes [5,6,7]. Consequently, a subsequent cleaning process is required [8,9]. The quality of the cleanliness of the metal surface is critical in the industrial environment. Poor cleanliness levels lead to a decrease in adhesion, which reduces reliability [10,11]. Several detection methods have been developed to investigate the reliability of wire bonding, soldering, or direct packing. Contamination often occurs during soldering processes when using fluxes, which can flow over and wet the surface. These fluxes contain organic acids and alcohol, which volatilize due to the increased temperature during the soldering process and react with the residue of organic acids, producing a salt [12,13,14,15] and initiating corrosion processes [16,17].

Traditionally, methods such as X-ray photoelectron spectroscopy (XPS) and Auger electron spectroscopy (AES) are used to determine cleanliness [18,19]. Hsieh et al. [20] investigated the effectiveness of plasma cleaning in removing contaminants from the surface of copper lead frames by using Ar and Ar/H_2_ gases. They provide detailed information, but the method is time-consuming and expensive [6]. Hyperspectral imaging is a technique that combines optical spectroscopy with imaging [21,22]. It extends the capability of standard spectroscopy by adding spatial dimensions (*x* and *y* coordinates) to the spectral information (*λ*) simultaneously [23,24,25]. As a result, hyperspectral imaging produces a 3D data structure (*x*, *y*, *λ*) known as a hypercube [26,27,28]. Such additional spatial information improves accuracy. Furthermore, hyperspectral imaging is a rapid, non-destructive, and robust approach, making it suitable for applications in many areas, such as food and pharmaceutical production, agriculture, military, and medicine [24,29,30,31]. Copper cleanliness classification for industrial purposes requires high lateral resolution [32]. Several studies have focused on the contamination of electronic direct bonded copper in visible (Vis), near-infrared (NIR), and ultraviolet (UV) regions [33]. Hyperspectral imaging and diffuse reflectance spectroscopy have been used to estimate the cleanliness, contamination, and qualification of thin films in different fields [4,5,6,34,35]. Stiedl et al. [5] detected and characterized the oxide layer thicknesses on metallic copper samples by using hyperspectral imaging in the UV–Vis spectral range. Al Ktash et al. [36,37] developed UV hyperspectral imaging for in-line industrial applications; they were able to precisely characterize oxide layer thicknesses and copper states on direct bonded copper by UV hyperspectral imaging [4]. Englert et al. [8] were able to identify the level of cleanliness on copper surfaces by using hyperspectral imaging to quantify organic contaminants on copper surfaces for electronic applications in the Vis/NIR region.

Hyperspectral imaging produces a large number of data, making it necessary to utilize multivariate data analysis techniques such as principal component analysis (PCA) and partial least squares regression (PLS-R) [38,39,40]. A combination of PCA and Bayesian discriminant analysis (DA) enables the classification of data and the additional deduction of model-related quality parameters [41]. PCA-DA is effective in classifying and visualizing groups within data clusters, while PLS-R is useful in evaluating the robustness of models in a quantitative way. In order to effectively analyze and interpret the results of high-resolution hyperspectral imaging, combining these three techniques is often required [42,43,44,45].

In the present work, hyperspectral imaging in the UV range was utilized to determine the cleanliness of direct bonded copper surfaces. The focus was on a specific cleaning step, using a defined reagent mixture to develop a predictive model capable of determining the optimal sonication time. Reducing the cleaning time while maintaining quality allows for cost savings in production by enabling higher throughput in a shorter amount of time. For this purpose, an existing pushbroom imaging setup [37] was augmented with a deuterium lamp. Compared with a xenon arc lamp, the deuterium lamp provides higher intensity in the short-wavelength UV range [31,37]. This range is particularly sensitive to organic compounds. Consequently, the UV range enables the prediction of cleanliness levels and, with appropriate modelling, also allows for the measurement of copper oxide layer thickness. Consequently, the cleanliness of soldered copper substrates and organic contamination surfaces is quantified by multivariate data analysis. As a result, the specific levels of cleanliness can be determined, reducing costs related to time, energy, and resources.

## 2. Materials and Methods

### 2.1. Samples

A total of 44 direct bonded copper Curamik^®^ Power substrates (Rogers Corporation, Chandler, AZ, USA) were used. These substrates had dimensions of 21.0 mm × 21.0 mm × 1.1 mm. The solder paste F360 SnAg 3.5 (Hereaus, Hanau, Germany) was printed on the sample’s surface with dimensions of 18.0 mm × 3.0 mm × 0.2 mm. The samples were soldered at 240 °C for 4 min in a nitrogen atmosphere, cooled down to 25 °C and directly cleaned. For each sample set (Set0–Set10), four direct bonded copper substrates were soldered and cleaned. The sample set was split into two groups: model building and model prediction (see Figure 1).

### 2.2. The Wet Chemical Cleaning Process

All samples were immersed and sonicated with a mixture of the cleaning agents: 30% Vigon A200 (Zestron, Ingolstadt, Germany) and 70% deionized water. During the cleaning process, the temperature of the cleaning agent was maintained at 50 °C. In the next step, the samples were rinsed with deionized water for 1 min and air-dried at room temperature for 2 h. After drying, the samples were directly measured to avoid oxidation. Only the sonication duration was varied to produce different levels of cleanliness (Table 1). Set0 was left in the initial conditions.

### 2.3. UV Hyperspectral Imaging

The hyperspectral imaging setup was improved compared with previous studies [4,31,36] and complemented by an SL3 deuterium lamp (24 V, 65.04 W; StellarNet Inc., Tampa, FL, USA). The spectral radiance for a deuterium lamp and a xenon-arc lamp is shown in Figure A1. The reflectance of the direct bonded copper samples was recorded in the UV range from 225 to 400 nm. Figure 2a shows a schematic of the hyperspectral imaging setup. The UV line scanner (pushbroom) consisted of a back-illuminated CCD camera (Apogee Alta F47: Compact; inno-spec GmbH, Nürnberg, Germany) connected to a spectrograph (RS 50–1938; inno-spec GmbH, Nürnberg, Germany), with a slit width of 30 µm. The camera’s resolution was 1024 pixels × 1024 pixels (spatial × spectral), and the pixel size was 13 µm × 13 µm. The integration time was 300 ms. The samples were placed on a black conveyor belt (700 mm × 215 mm × 60 mm; Dobot Magician, Shenzhen Yuejiang Technology Co., Ltd., Shenzhen, China), moving at a constant speed of 0.15 mm/s.

Figure 2b illustrates the principle of continuous line-by-line spectral data collection. The data result in a laterally resolved (*x*, *y*) 2D image, as shown in Figure 2c, where each location contains an additional spectroscopic dimension (*λ*), as shown in Figure 2d. Thus, a 3D data matrix (hypercube) was recorded [36]. The UV hyperspectral imaging data were obtained by SI-Cap-GB version V3.3.x.0 software (inno-spec GmbH, Nürnberg, Germany). The reflectance was calculated by recording I_reference_ and I_dark_ according to the radiometric calibration [36,42]:(1)$$\mathrm{Reflectance}\;=\frac{R}{R_{0}}=\frac{I_{\mathrm{sample}}\;-\;I_{\mathrm{dark}}}{I_{\mathrm{reference}}\;-\;I_{\mathrm{dark}}}$$

R and R_0_ represent the reflected intensities from the sample and a reference material with high reflectivity, respectively, in this case, PTFE (polytetrafluoroethylene). The intensity of the original image is denoted by I_sample_. Correspondingly, the intensity of the dark current image is denoted by I_dark_, and the intensity of the PTFE image is I_reference_. To enhance the spectral bands, the negative decadic logarithm of the ratio R/R_0_ is calculated as −log(R/R_0_), thereby expressing the data in terms of absorbance.

Prior to use, the pushbroom imager was calibrated with a high-precision laboratory spectrometer (Lambda 1050; PerkinElmer, Inc., Waltham, MA, USA), mapping the 1024 pixels to the spectral range of 225–400 nm. Each pixel corresponds to a specific wavelength, thereby defining the spectrum. Consequently, the spectrum was represented by 1024 discrete variables, which were analyzed by using multivariate techniques. This configuration yields a mathematical resolution of Δ*λ* = 0.2 nm. However, the actual spectral resolution is determined by the optics, particularly the spectrograph grating, and is typically lower.

### 2.4. Data Processing and Multivariate Data Analysis

Evince version 2.7.11 (Prediktera AB, Umeå, Sweden) was used to extract the hyperspectral data matrices. Ten average spectra for each sample (thirty for each set) were acquired in the region of interest (Figure 1). Each average spectrum consisted of 200 single spectra. Each spectrum contained 1024 variables/pixels (wavelength axis). In total, 800 spectra (equivalent to 800 pixels) for each sample type were the basis for the multivariate data analysis (8.800 spectra for all samples). The extracted UV spectra were used as input for the subsequent PCA-DA and PLS-R. Multivariate data analysis was performed with “The Unscrambler X 10.5” (Camo Analytics AS, Oslo, Norway). All spectra were preprocessed by using Savitzky–Golay smoothing (21 points, symmetric, 2nd polynomial order). PCA was performed with mean centering, full cross-validation, and the NIPALS algorithm to distinguish among the cleanliness levels. The PCA result was subsequently combined with a DA (quadratic, 2 PCs) to determine the model’s quality parameters. Accuracy, sensitivity, specificity, and precision were calculated based on the confusion matrix terminology [45,46].

For the PLS-R model, mean centering, full cross-validation, and the Kernel algorithm were applied. Three direct bonded copper samples of each type were used to develop the PLS-R model, and one sample of each sample type was used to test the final PLS-R model’s performance.

## 3. Results and Discussion

A sample set of 44 direct bonded copper samples (11 sample types, with 4 samples per type) were investigated by using hyperspectral imaging in the UV region (225–400 nm). This sample set was created to develop a model for predicting the cleanliness level of direct bonded copper based on the sonication duration time as a Y-reference. This Y-reference was selected to illustrate, using a simple and cost-effective example, how process steps can be easily optimized by using UV hyperspectral imaging as a method for quality control. The previous study by Englert et al. used XPS on single spots (two points per sample) to correlate with Vis/NIR hyperspectral imaging [8]. Figure 3 shows the averaged and preprocessed reflectance spectra of each sample type with linear baseline correction. The vertical shift of the averaged spectra allows for the assignment of different sonication time steps in the cleaning process from low (set0) to high (set10). Sample set0 represents the direct bonded copper in its initial condition without any cleaning step. The other samples underwent a progressively longer cleaning process, as described in Section 2.2. Weak shoulders at 241 nm, 265 nm, 362 nm, and 385 nm can be assigned to a thin layer of Cu_2_O and CuO [4,5]. The cleaner the sample, the less pronounced is the overall reflectance. The most dominant band is identified approximately at 300 nm and assigned to copper (Cu^0^) [4]. As the cleanliness process proceeds, the band at 300 nm narrows. This effect can be attributed to the removal of organic compounds and copper oxides [8].

A PCA model with full cross-validation was calculated for the 330 average spectra of all samples. The averaged spectra were used to minimize the heterogeneity among samples and enhance correlation. Figure 4a shows the scores plot of PC1 (99.75%) and PC2 (0.25%). The first two PCs explain nearly 100.0% of the total variance. The scores of the different sample sets are distinguishable. Each direct bonded copper sample set and its corresponding cleanliness level appear nearly distinct. As the cleanliness level increases, the overlap of the sample sets also increases. In contrast, the variance within a sample set decreases due to reduced organic contamination and oxidation states. Cleaner spectra at each pixel lead to cleaner areas, thereby accumulating similarities among the direct bonded copper substrates.

From positive to negative scores on PC1, the samples are arranged in order of cleanliness process level from the initial condition (set0) to the cleanest sample (set10). PC2 amplifies this effect and separates set1, set2, set3, and set4 (negative scores) from the others (positive scores).

The loading plots for PC1 and PC2 are given in Figure 4b,c. The shapes of PC1 and PC2 resembles copper with its different oxidation states (Cu^0^, Cu_2_O, and CuO). The most dominant bands are pronounced at 260 nm and 300 nm, related to the presence of Cu^0^, Cu_2_O, and CuO (Figure 3) [4,5,8].

In order to validate the quality of the PCA model, the scores of the first two PCs were used to perform a quadratic discriminant analysis (QDA). Table 2 presents the confusion matrix for all direct bonded copper sample sets (set0, initial conditions; set10, the cleanest). The confusion matrix provides a description of the classification model’s capability [36,47]. Overall accuracy reaches 70.6%, with sensitivity at 76.9%, specificity at 73.0%, and precision at 78.2% for the spectra. As the cleanliness level increases, misclassifications (highlighted in light gray) among the sample sets increase, while correctly classified spectra (dark-highlighted diagonal) decrease. The dark highlighted diagonal elements represent the accordance of the predicted and actual values. Misclassification can be attributed to the increasing similarity between the direct bonded copper substrates and the cleanliness level. Quality control aims to ensure consistent product conditions, as demonstrated here.

In the next step, a PLS-R model was developed by using a calibration set of *n* = 33 samples, employing two factors, the Kernel algorithm, and segmented full cross-validation. A prediction sample set was then used to evaluate the PLS-R model’s performance through external validation, assessing its predictive ability. The prediction sample set comprised 11 samples with varying cleanliness levels, including 1 sample per sonication time (Table 1). Figure 5a illustrates the correlation between the reference and predicted spectra from cross-validation. The regression function (red diagonal) is represented by *y* = 0.93*x* + 0.35. The model’s performance in calibration and validation was evaluated by using the coefficient of determination (*R*^2^) for the calibration (*R*^2^_c_ = 0.929) and segmented full cross-validation (*R*^2^_cv_ = 0.928). The root-mean-square errors of calibration (RMSEC = 0.83) and cross-validation (RMSECV = 0.85) provide further insights into model performance. Figure 5b displays the regression coefficients for the two-factor model. Once again, the most impacts of the regression coefficients are attributed to copper material (Cu^0^) and its oxidation states (Cu_2_O and CuO) [4,5,8].

In order to evaluate the PLS-R models, the remaining 11 samples were used to assess the model’s performance. The cleanliness level of each pixel (single spectra) in the region of interest was predicted. Figure 6 shows the resulting distribution map, where pixels represent the cleanliness of the direct bonded copper sample set, ranging from sample set0 (initial condition) to the cleanest sample (set10). The color bar denotes reflectance intensity, ranging from low (blue) to high (red). At the sample corners (high-intensity parts), organic contamination from fluxes in the solder paste is prominent [8]. The middle area of sample set0 exhibits artifacts due to high reflectivity at the corners, where signals from the corners overlay those from the middle area. From sample set1 to sample set10, the intensity at the corner diminishes, leadings to more homogenous sample surfaces. Sample set10 shows no high-reflectivity pixels, indicating that a 10 min sonication time is sufficient to achieve satisfying cleanliness results.

UV hyperspectral imaging effectively distinguishes different cleanliness levels among the samples and for each pixel individually. Moreover, the results demonstrate that the UV hyperspectral imaging model is highly effective for predicting cleanliness procedures in a rapid, real-time, and cost-effective manner. Only two factors are necessary to achieve a model with high *R*^2^ and low RMSE.

## 4. Conclusions

The cleanliness of direct bonded copper poses a significant challenge in electronics manufacturing quality assurance. UV hyperspectral imaging in the wavelength range of 225–400 nm was employed for non-destructive evaluation of cleanliness levels, exemplifying optimization in a production step. Therefore, the sonication time was correlated with UV hyperspectral imaging. A pushbroom imager-based UV hyperspectral imaging setup, coupled with a deuterium lamp, facilitated lateral measurements of copper surfaces. A total of 44 samples were analyzed to develop multivariate models for characterizing and predicting cleanliness levels.

The PCA model effectively differentiated all direct bonded copper sets by using the first two principal components according to the sonication time, which correlated to the cleanliness levels. The PLS-R model (*n* = 33) calculated with two factors yielded high *R*^2^ (*R*^2^_c_ = 0.929 and *R*^2^_cv_ =0.928) and low errors (RMSEC = 0.838 and RMSECV = 0.849). The remaining 11 samples were used to successfully test the PLS-R model’s performance at each pixel. As a consequence, 10 min sonication time and occasional at-line spot checks with UV hyperspectral imaging suffice to achieve satisfying cleanliness results. The UV hyperspectral imaging prototype offers distinct advantages in terms of high spatial and spectral resolution, as well as rapid data acquisition speed under laboratory conditions.

The utilization of UV hyperspectral imaging combined with multivariate modeling exemplifies a promising approach to optimizing the sonication duration. This method provides practical benefits for process applications and potential commercialization in an industrial context. It is adaptable to various manufacturing steps involving direct bonded copper substrates, enabling the at-line or in-line quality control of cleanliness during electrical device production. Leveraging the findings and data from this study can lead to the development of systems meeting industrial process requirements, thereby reducing waste and costs. In our future studies, a combination of both illumination types, deuterium and xenon arc, will be utilized.

## Figures and Tables

**Figure 1 sensors-24-04680-f001:**
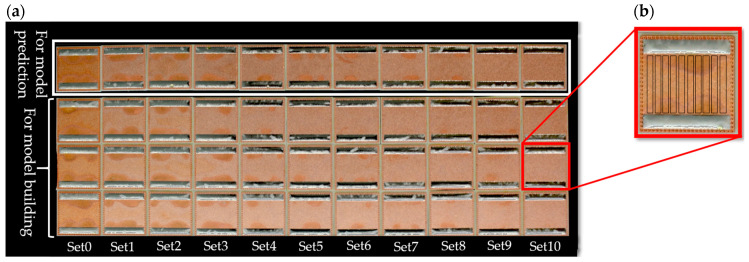
(**a**) Direct bonded copper Curamik^®^ Power substrates. (**b**) Magnified image of a direct bonded copper sample with the regions of interest marked by black rectangles. The average spectra were extracted from these regions. A total of 44 direct bonded copper samples were divided into two groups, where 33 samples were used for model building and 11 samples were used for model evaluation.

**Figure 2 sensors-24-04680-f002:**
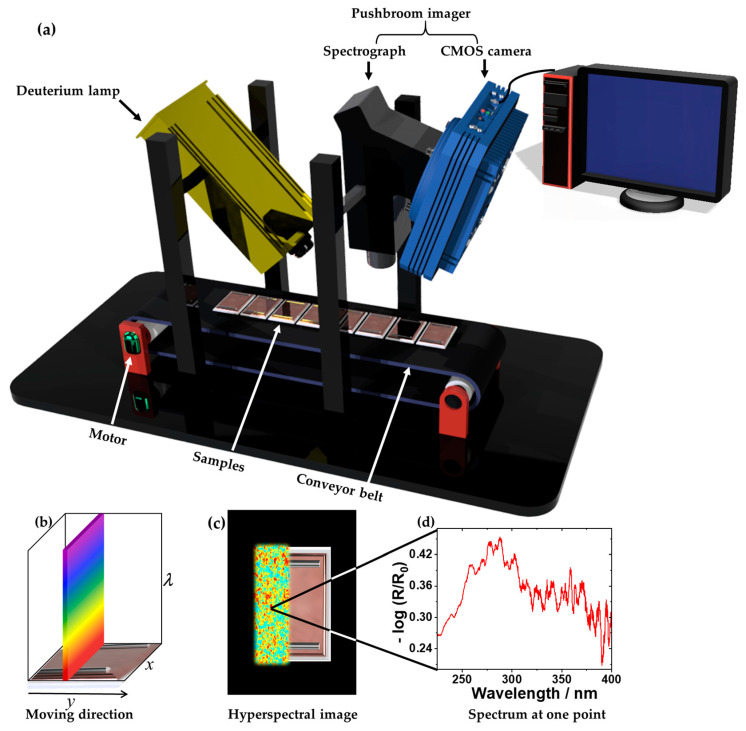
(**a**) A schematic setup of the UV hyperspectral imaging prototype based on the pushbroom concept. (**b**) The principle of the pushbroom imaging scan on the example of a direct bonded copper sample. (**c**) A hyperspectral image of a scanned sample with a UV spectrum at each pixel. (**d**) The extracted UV spectrum (225–400 nm) from the image given in (**c**).

**Figure 3 sensors-24-04680-f003:**
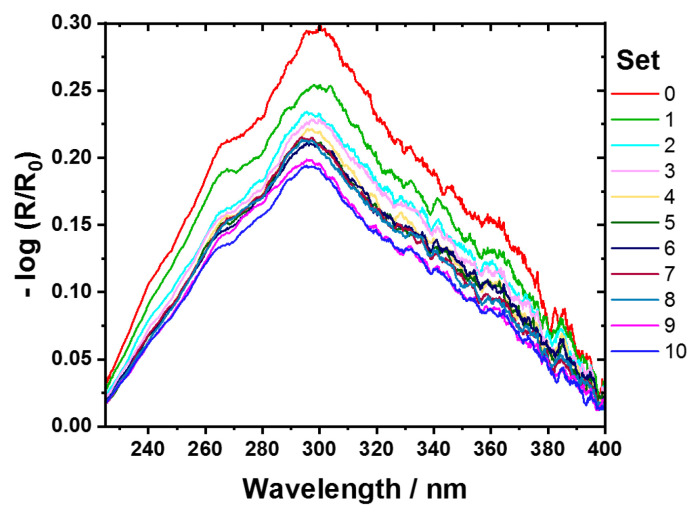
UV reflectance average spectra of direct bonded copper (225–400 nm) in absorbance. From copper sample set0 in its initial conditions (red) to the cleanest sample, set10 (blue). Savitzky–Golay smoothing (21 points, symmetric, 2nd polynomial order) was employed for data preprocessing.

**Figure 4 sensors-24-04680-f004:**
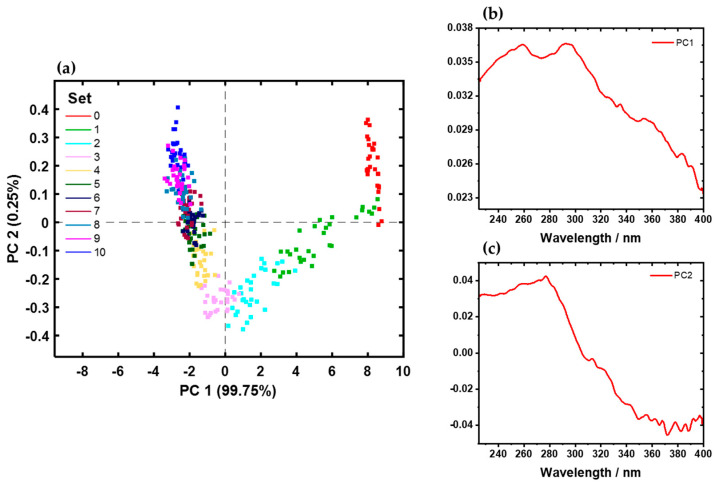
PCA model with (**a**) scores in PC1 (99.75%) against PC2 (0.25%) for all 44 direct bonded copper samples and (**b**,**c**) PC1 (99.75%) and PC2 (0.25%) loadings plots, respectively. The copper sample from initial conditions (set0; red) to the cleanest sample (set10; blue) are shown in (**a**).

**Figure 5 sensors-24-04680-f005:**
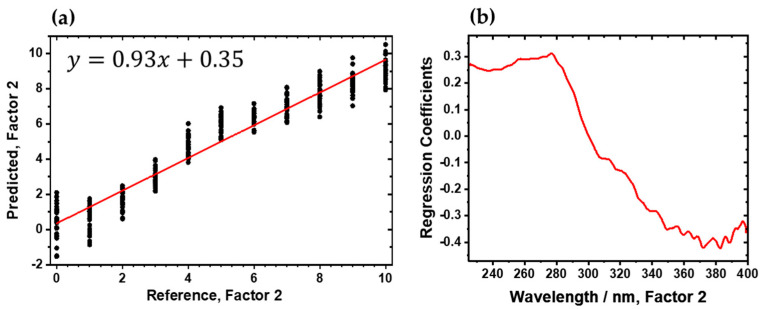
Two-factor PLS-R model for cleanliness process of direct bonded copper in UV region (225–400 nm). (**a**) Predicted vs. reference of UV hyperspectral imaging of cross-validation. (**b**) Regression coefficients of two-factor model.

**Figure 6 sensors-24-04680-f006:**
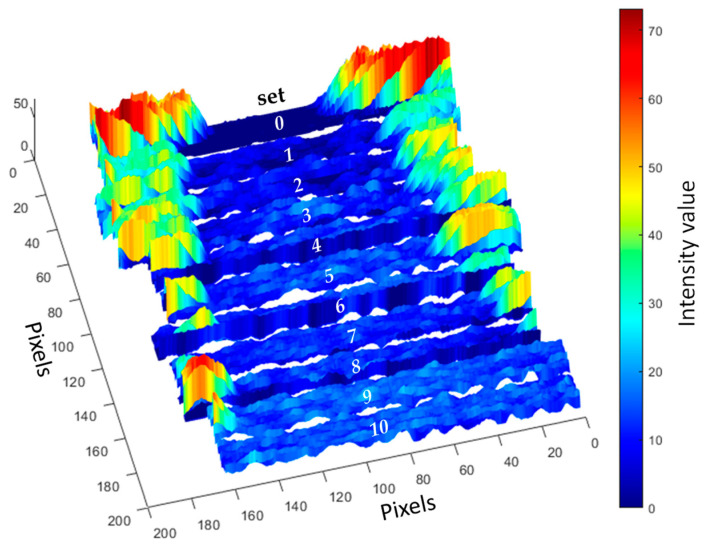
Distribution map predicted from the two-factor PLS-R model of the UV hyperspectral imaging data. The cleanliness process for each pixel of samples from sample set0 to sample set10 was calculated for model prediction.

**Table 1 sensors-24-04680-t001:** Sample preparation protocol for direct bonded copper substrates.

**Sample Set**	Set0	Set1	Set2	Set3	Set4	Set5	Set6	Set7	Set8	Set9	Set10
**Sonication duration time (min)**	0	1	2	3	4	5	6	7	8	9	10

**Table 2 sensors-24-04680-t002:** Confusion matrix of the PCA-QDA model with 2 PCs projected (overall accuracy 70.6%). Correctly classified spectra are highlighted in dark. Misclassifications are highlighted in light gray.

Actual												
**Predicted**	Sample set	0	1	2	3	4	5	6	7	8	9	10
	0	30	2	0	0	0	0	0	0	0	0	0
	1	0	26	2	0	0	0	0	0	0	0	0
	2	0	2	26	1	0	0	0	0	0	0	0
	3	0	0	2	27	0	0	0	0	0	0	0
	4	0	0	0	2	25	5	0	0	0	0	0
	5	0	0	0	0	4	10	6	3	0	0	0
	6	0	0	0	0	1	11	19	5	2	0	0
	7	0	0	0	0	0	3	5	16	6	1	0
	8	0	0	0	0	0	0	0	4	12	6	1
	9	0	0	0	0	0	0	0	2	7	18	5
	10	0	0	0	0	0	0	0	0	3	5	24

## Data Availability

The raw/processed data required to reproduce these findings cannot be shared at this time as the data also form part of an ongoing Ph.D. thesis.

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
