# Peer review of "Rapid Detection of Cleanliness on Direct Bonded Copper Substrate by Using UV Hyperspectral Imaging"

_sensors, 2024, doi:10.3390/s24144680_

Round 1

Reviewer 1 Report

Comments and Suggestions for Authors

The paper discusses the use of hyperspectral imaging in the UV to evaluate the cleanliness of copper samples by different sonication times. It presents a compelling argument for using a pushbroom imager in a web process where samples move along an assembly line. The data is analyzed using PCA and PLSR, both of which convincingly demonstrate that different levels of cleanliness can be accurately detected using UV imaging spectroscopy.

I suggest the following comments to further enhance the thoroughness and clarity of the manuscript:

-             What are the benefits of using a deuterium lamp compared to an XBO lamp?

-             Why didn’t you use both lamps at the same time?

-             In the description of Figure 3, add what kind of preprocessing you used for the spectra. Also, add Absorption on the y-axis instead of the equation.

-             In Figures 5 b and c, how much loadings explain the data? Is it the same as scores? Add the percent in the legend.

In conclusion, I found the paper well-written and structured, and I commend the authors for their work. Therefore, after considering my comments, I recommend it for publication in Sensors.

Author Response

Dear Reviewer 1,

please see the attached file.

Kind regards,

the authors

Reviewer 2 Report

Comments and Suggestions for Authors

The paper reads very well and all figures are properly design accepted as is

Author Response

Comment: The paper reads very well and all figures are properly design accepted as is.

Response to Reviewer 2:

Dear Reviewer 2,

Thank you very much for your time and effort in reading and evaluating our manuscript. We are very pleased that you have no suggestions for changes. Thank you and best regards,

The Authors

Reviewer 3 Report

Comments and Suggestions for Authors

This manuscript has some originality, the article is well structured and should be carefully revised grammatically. The manuscript is acceptable with modifications.

Comments on the Quality of English Language

This manuscript  should be carefully revised grammatically.

Author Response

Comment 1: This manuscript has some originality, the article is well structured and should be carefully revised grammatically. The manuscript is acceptable with modifications.

Response 1: 

Dear Reviewer 3,

Thank you very much for your time and effort in reading and evaluating our manuscript. We are very pleased that you have no suggestions for changes. Thank you and best regards,

The Authors

Comment 2: This manuscript  should be carefully revised grammatically.

Response 2: We revised the manuscript grammatically and changed some mistakes. Thank you very much!